# Feasibility and agreement between TTE and intracavitary ECG for PICC tip positioning in adult oncology patients: A single-centre exploratory study

Kai Xin[1], Mian Zhou[2], Yubiao Kang[3], Shanping Li[1], Chenjuan Wang[1], Lin Wang[1], Ling Yuan [1]*, Baorui Liu[1]*

1 Department of Oncology, Nanjing Drum Tower Hospital, Affiliated Hospital of Medical School, Nanjing University, Nanjing, China, 2 Department of Thoracic, Nanjing Drum Tower Hospital, Affiliated Hospital of Medical School, Nanjing University, Nanjing, China, 3 College of Nursing, Nanjing University of Chinese Medicine, Nanjing, China

☯ These authors contributed equally to this work.
* baoruiliu@nju.edu.cn (BL); yuanling@nju.edu.cn (LY)

## Abstract

### Background

ECG-guided tip localization of central venous access has been clinically proven to be highly accurate and safe. However, real-time tip localization in patients with atrial fibrillation remains a challenge. Transthoracic ultrasound catheter tip localization is noninvasive and real-time. However, no methodological or accuracy studies of its localization have been reported in the literature.

### Objective

To explore the accuracy of the TTE technique in localizing the tip position of central venous access devices (CVADs) and to establish a standardized TTE-guided catheter tip positioning procedure.

### Methods

This was an self-controlled trial. A total of 35 patients requiring PORT or PICC implantation participated in this study from April 2023 to March 2024. First, catheter positioning was performed under TTE guidance, and after determining the position of the catheter tip, the insertion depth and relative data were recorded. Subsequently, ECG-guided positioning was performed. The final position of the catheter was standardized by ECG.

**Data availability statement:** The minimal dataset underlying the findings of this study is provided within the Supporting Information files as a compressed archive (S1 File). All data have been anonymized to protect patient confidentiality. No individual-level identifiable information is included.

**Funding:** This work was supported by Nanjing Drum Tower Hospital (Grant No. 2023-A494).

**Competing interests:** The authors have declared that no competing interests exist.

## Results

A total of 35 patients successfully underwent CVAD insertion under the guidance of IC-ECG. Five patients were excluded from the per-protocol (PP) analysis because the catheter tip could not be visualized in the Apical Four Chamber View. Statistical analysis showed no significant difference between ECG-measured depth and TTE-measured depth (t = −1.405, p = 0.17). In the PP Bland–Altman analysis, the mean bias was −0.13 with 95% limits of agreement (LOA) from −1.10 to 0.83, indicating good agreement between the two methods. An additional intention-to-diagnose (ITD) analysis with extreme value imputation for the five missing cases yielded a mean bias of −0.34 and 95% LOA from −1.70 to 1.00, further confirming the robustness of the findings. The pain intensity score did not exceed 4 points for 34 patients, with only one patient reaching 5 points. The TTE operation time did not exceed 8 minutes for each patient.

## Conclusion

This study suggests that non-contrast TTE shows reasonable consistency with IC-ECG, with short operation time and acceptable patient discomfort. TTE may serve as a feasible, real-time, non-invasive alternative for patients in whom IC-ECG cannot be used.

## 1. Background

INS guidelines indicate [1] that the optimal position for CVADs is the lower 1/3 of the superior vena cava and Cavoatrial Junction (CAJ). It is recommended to use non-invasive catheter tip positioning techniques during catheter insertion, such as intra-cavitary electrocardiography (IC-ECG) or transthoracic echocardiography (TTE), and to discourage the use of pre-procedural external measurements or post-procedural X-ray verification methods. Deviation of the catheter tip from the optimal position can cause mechanical or chemical damage to blood vessels [2], emphasizing the crucial importance of confirming the catheter tip position.

Clinical practice often relies on postoperative chest X-rays or CT scans to determine the catheter tip position. However, these methods lack real-time applicability, and repeated postoperative catheter adjustments increase patient radiation exposure and discomfort. Additionally, the catheter tip position for implanted infusion ports requires intraoperative confirmation, a condition not available in all research centers.

Transesophageal echocardiography (TEE) is the most accurate positioning method, but it is invasive and challenging to use routinely in most clinical scenarios [3]. Digital subtraction angiography (DSA) achieves a high success rate in catheter tip positioning, but it is not recommended due to exposure to ionizing radiation during the procedure and its high cost [4]. IC-ECG positioning technology has high accuracy and is radiation-free, but it relies on the interpretation of electrocardiac signals [5] and is not suitable for patients without stable P-waves or with unstable P-waves [6]. TTE

is a non-invasive technique that utilizes the unique physical properties of ultrasound to examine the anatomical structures and functional status of the heart and major vessels. The examination modes of echocardiography typically involve 2D imaging, with additional options such as M-mode, color Doppler mode, spectral Doppler mode, tissue Doppler mode, etc., depending on the requirements. The combination of multiple modes provides more comprehensive information [7]. Catheter tip positioning in the atrium can usually be achieved with 2D ultrasound, and real-time and safe TTE technology has been proven feasible in domestic and international studies, showing promising prospects [8–10]. However, research on using TTE for positioning the catheter tip of CVADs mainly focused on improving the sensitivity of this technique, such as "microbubble tests", lacking emphasis on standardized operating procedures and reliability data, leading to a deficiency in evidence-based medicine [11]. This limitation hampers its clinical application.

Previous studies have primarily focused on contrast-enhanced transthoracic echocardiography (CE-TTE) or transe-sophageal echocardiography (TEE) for catheter tip localization [12], both of which have demonstrated high accuracy but are either invasive or require contrast injection that may limit routine use. In contrast, evidence regarding the accuracy and standardization of non-contrast TTE remains scarce. Moreover, no prior study has systematically validated non-contrast TTE against intracavitary electrocardiography (IC-ECG), which is widely recognized as a safe, radiation-free, and clinically practical reference standard. Our study therefore addresses this important gap by prospectively evaluating the performance of non-contrast TTE in comparison with IC-ECG, and by proposing a standardized, real-time TTE-guided procedure for central venous access device (CVAD) tip localization.

The purpose of this study was to investigate the accuracy and safety of non-contrast TTE-guided catheter tip positioning for central venous access devices and to establish the methodology of the technique.

## 2. Materials and methods

### 2.1 Study design

This prospective, self-controlled trial was registered with the Chinese Clinical Trial Registry (registration number: ChiCTR2300074384; registration date: August 4, 2023). https://www.chictr.org.cn/bin/project/edit?pid=191335. The Institutional Review Board of our hospital approved this study (approval no.2023-115-01) and informed consent was obtained from all patients.

### 2.2 Study patients

Patients were recruited from April 2023 to March 2024 at an oncology treatment center. The research process is shown in Fig 1. The investigators communicated with potential patients who met the inclusion criteria and enrolled them after obtaining written informed consent. The inclusion criteria were: 1) Malignant tumors requiring PICC and PORT catheters for cyclic infusion of chemotherapy drugs. 2) Patients aged >18 years. 3) Consent to participate in this study and sign the informed consent form for this study. The exclusion criteria were: 1) Patients who have recently undergone major chest surgery and cannot tolerate ultrasound monitoring. 2) Patients with contraindications to central venous catheter place-ment. 3) Suffering from thrombophilia. 4) Patients with abnormal coagulation function.

The primary outcome indicator of the study was catheter tip position. Secondary outcome indicators included infection, operation time, and patient chest wall pain.

### 2.3 Intervention and control conditions

All CVADs placements occurred in a class 100 operating suite in an oncology center at a university hospital. All patients signed an informed consent form to participate in this study. All catheters were implanted by the same operator and ultra-sound examinations were performed by a physician with relevant qualifications. TTE images from different views (using the Philips Sparq System with S4-2 phased-array probe) were captured. All catheter insertion procedures were conducted according to the standard protocols, with slight modifications, as follows:

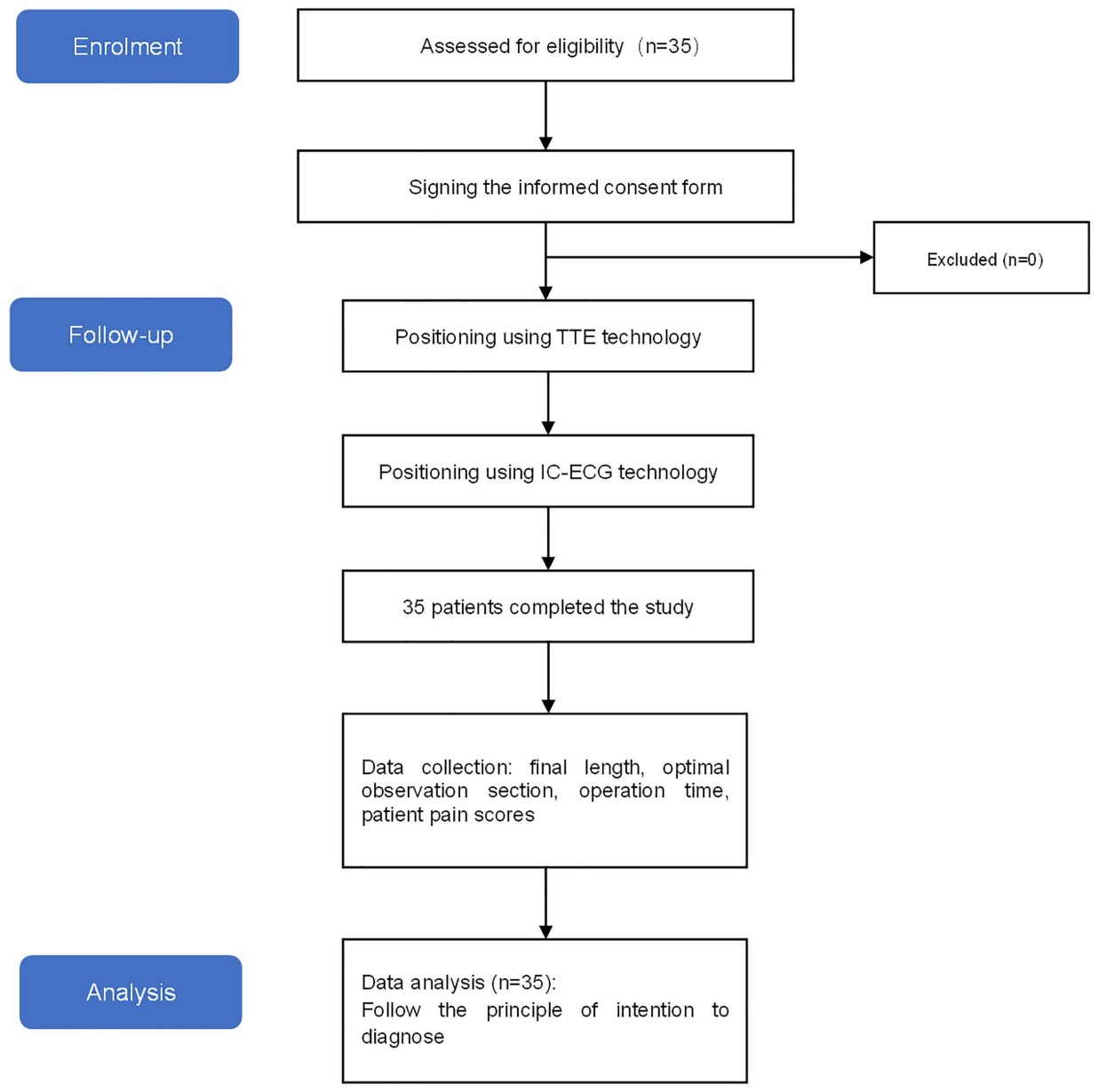

**Fig 1. The flowchart of this research.**

- The patient was placed in a supine position with the arm abducted at 90°. After disinfection and draping, local anesthesia was administered, and under ultrasound guidance, the puncture was performed to introduce the catheter.

- Bedside echocardiography was performed and TTE images were captured.

- The ultrasound probe was placed in the *Parasternal Short Axis View, Apical Four Chamber View, and Subxiphoid Biatrial View*. The operator advanced the catheter under continuous ultrasound visualization until the catheter tip was visualized on TTE. The ultrasound probe was continually adjusted for optimal viewing position.

- The distance from the catheter tip to the interventricular septum was measured as VL in the *Apical Four Chamber View*. The predicted catheter insertion depth under the TTE guidance was calculated as X(tte)= X(np) − VL.

- Relevant measurements were recorded prospectively.

- The IC-ECG technique was subsequently used to locate the catheter tip position. The electrocardiogram waveform was observed for a negative P-wave.

- The catheter was then retracted until the electrocardiogram displayed the positive peak of the P-wave, and the final actual catheter insertion depth X(ecg) was recorded.

- Relevant measurements were recorded prospectively.

## 2.4  Variables and measurement methods

- Optimal length/final length: IC-ECG <u>was</u> used to guide catheter implantation, and when the negative P-wave appeared, the catheter was slowly retracted until the highest positive P-wave appears.

- Optimal observation section: The ultrasound probe <u>was</u> placed in the *Parasternal Short Axis View*, *Apical Four Chamber View* and *Subxiphoid Biatrial View*, and the observation effect under the three sections <u>was</u> compared, and the section in which most patients could observe the relevant anatomical position and the catheter tip is the optimal observation section.

- Operation time: In this study, operation time is defined as the time spent on the application of the relevant technology. IC-ECG group time is defined as the time spent from the time when the operator changes the intracavitary ECG waveform to the time when the operator decides that the catheter is in the optimal position. TTE group time is defined as the time from the time when the operator starts to use the ultrasound probe in the optimal observation plane to the time when the operator decides that the tip of the catheter is in the optimal position.

- Patient pain scores: at the completion of the TTE technique application, patients were assessed using a numerical rating scale (NRS). As the IC-ECG technique is generally considered painless, only discomfort related to the TTE procedure was assessed. Patients were specifically informed that pain related to venous puncture was not included in the assessment.

## 2.5  Statistical analysis

The Horos was used for storing and analyzing DICOM image data. The Excel was used for data inputting and storage. All statistical analyses were performed using SPSS version 20.0 (IBM, USA). The normality of continuous variables was assessed using the Shapiro–Wilk test. Normally distributed paired continuous variables were expressed as mean ± SD and compared using paired-samples t-test. Non-normally distributed variables were expressed as median (IQR) and compared using the Wilcoxon signed-rank test. Categorical variables were presented as counts and percentages and compared using chi-square or Fisher's exact tests, as appropriate. Ordinal data were analyzed using the rank-sum test. A p-value < 0.05 was considered statistically significant. To address missing data, five patients in whom TTE measurements could not be obtained were included in an intention-to-diagnose (ITD) analysis. For these patients, extreme values were imputed to represent a worst-case scenario, in order to test the robustness of the agreement between TTE and ECG.

## 2.6  Ethical considerations

This study was approved by the ethics committees of the hospital (Ethics number: 2023-115-01). The researchers informed the patients in detail about the purpose and process of the study, answered the patients' questions, and

emphasized that they could withdraw from the study at any time and that withdrawal would not affect their treatment. In addition, the researchers assured the patients that their personal information would be kept confidential and that the data collected would be used for academic purposes only.

## 3. Result

### 3.1 General information

A total of 35 patients were enrolled (Table 1). Of these, 30 were included in the per-protocol (PP) analysis, while all 35 were included in an intention-to-diagnose (ITD) analysis, in which the five non-visualized cases were classified as 'non-diagnostic', due to the inability to observe the catheter tip position in the *Apical Four Chamber View*. This dual analysis provides a more comprehensive assessment of TTE performance.

### 3.2 Main indicators

**3.2.1 Observation view with TTE.** In 30 cases (85.7%), the catheter tip was visible in the right atrium on *Apical Four Chamber View*, compared to 10 cases (28.6%) on *Parasternal Short Axis View*, and 4 cases (11.4%) on *Subxiphoid Biatrial View*. There was a statistically significant difference in visibility rates among the three views ($\chi 2 = 43.502$, $p < 0.05$). Five cases were excluded due to inability to visualize the catheter tip by TTE. These patients tended to have higher BMI, thicker chest walls, or a history of chronic obstructive pulmonary disease (COPD), which may have limited the acoustic window and contributed to poor visualization. (Table 2)

**3.2.2 Consistency analysis.** In this study, we compared the mean values of the depth of catheter implantation guided by the two techniques. As shown in Table 3, comparison of the mean values of catheter tip depth measured by TTE and ECG revealed no significant difference between the two techniques ($t = -1.405$, $p = 0.17$). To further evaluate the agreement between methods, Bland–Altman analysis was performed. In the per-protocol (PP) analysis (Fig 2), the mean bias was −0.13, with 95% limits of agreement (LOA) ranging from −1.10 to 0.83, indicating good agreement without evidence of systematic bias. In the intention-to-diagnose (ITD) analysis, which included five additional patients with imputed extreme values (Fig 3), the mean bias was −0.34, with 95% LOA ranging from −1.70 to 1.00. Although the limits were slightly wider compared with the PP analysis, the overall agreement remained robust and clinically acceptable.

**Table 1. General patient information.**

|  | classification | N = 35 | ratio (%) |
|---|---|---|---|
| Age | 30-39 | 1 | 2.7 |
|  | 40-49 | 5 | 13.5 |
|  | 50-59 | 12 | 35.1 |
|  | ≥60 | 17 | 48.6 |
| Sex | Male | 12 | 32.4 |
|  | Female | 23 | 67.6 |
| Type | PICC | 29 | 83.8 |
|  | PORT | 6 | 16.2 |
| Diagnosis | Esophageal cancer | 6 | 18.9 |
|  | Breast cancer | 14 | 40.5 |
|  | Gastric cancer | 9 | 24.3 |
|  | Colorectal cancer | 3 | 8.1 |
|  | melanoma | 2 | 5.4 |
|  | Lung cancer | 1 | 2.7 |

**Table 2. The tip visibility rates from different cardiac ultrasound views.**

| Group | Invisible | Visible | Ratio (%) | $\chi^2$ | p |
|---|---|---|---|---|---|
| Parasternal Short Axis View | 25 | 10 | 28.6 | 43.502[a] | <0.05 |
| Apical Four Chamber View | 5 | 30 | 85.7 | | |
| Subxiphoid Biatrial View | 31 | 4 | 11.4 | | |

[a]Pearson's chi-square test.

**Table 3. Catheter Insertion Depth with IC-ECG and TTE (PPS).**

| | N | MEANA±STD | 95%CI | MIN~MAX | t | P |
|---|---|---|---|---|---|---|
| IC-ECG | 30(0) | 39.73±2.89 | 28.86~40.68 | 35.00~47.00 | −1.405[a] | 0.17 |
| TTE | 30(0) | 39.61±3.01 | 38.74~40.66 | 34.21~47.07 | | |
| TTE – IC-ECG | 30(0) | −0.13±0.49 | −0.31~0.06 | −1.67~1.08 | | |

SD, standard deviation.

[a]t-Test for paired samples.

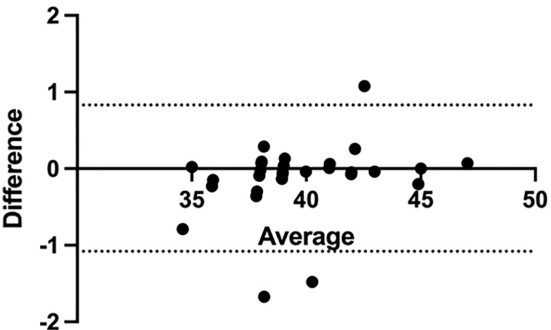

**Fig 2. Bland–Altman plot comparing TTE and ECG methods in the per-protocol (PP).**

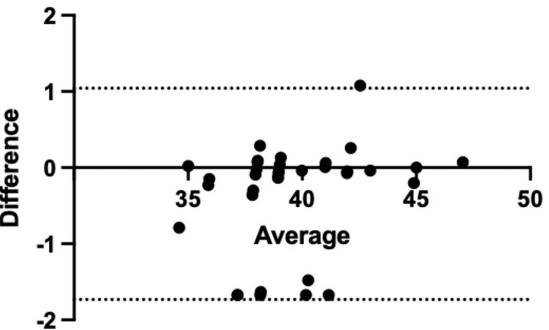

**Fig 3. Bland–Altman plot comparing TTE and ECG methods in the intention-to-diagnose (ITD) analysis with extreme value imputation.**

Fig 2 shows the differences between transthoracic echocardiography (TTE) and intracavitary electrocardiography (ECG) measurements against their averages. The solid line represents the mean bias (−0.13), while the dotted lines indicate the 95% limits of agreement (−1.10 to 0.83). Most data points fall within these limits, demonstrating good agreement between the two methods without evidence of systematic bias.

This sensitivity analysis included five additional patients with imputed extreme values in whom TTE measurements were not obtainable. The mean bias was −0.34, and the 95% limits of agreement ranged from −1.70 to 1.00. Although the limits were slightly wider than those in the PP analysis, the overall agreement between TTE and ECG remained robust and clinically acceptable.

**3.2.3 Operation time.** The operation time was 2.7 minutes in the ECG group and 4.3 minutes in the TTE group. There was a statistically significant difference in operation time between the two groups ($p < 0.05$). Although TTE required slightly longer operation time, both procedures were completed within a short and clinically acceptable duration. (Table 4)

**3.2.4 Pain intensity score in TTE group.** Patients in the TTE group had pain scores ranging from 1 to 4, with a mean of 2.1 ± 1.1. (Table 5)

## 4. Discussion

### 4.1 *Apical Four Chamber View* is the best observation view

Common cardiac ultrasound views include: (1) *Left Ventricular Long-Axis View*: This slice displays the long-axis view from the base to the apex of the heart, primarily used to assess overall left ventricular motion and examine the movement of the mitral and aortic valves; (2) *Parasternal Short Axis View*: Obtained by clockwise rotation of the left ventricular long-axis view, it observes the motion of the right atrium, tricuspid valve, right ventricular outflow tract, pulmonary artery, and measures pulmonary artery pressure; (3) *Left ventricular short-axis view*: Observes the motion of different levels of left ventricular wall muscles; (4) *Apical Four Chamber View*: Provides an overall view of the heart's four chambers and three valves' motion; (5) *Subxiphoid Biatrial View*: Observes the motion of the left and right atria, commonly used for the assessment of atrial septal defects; (6) *Suprasternal Aortic Arch Long-Axis View*: Displays the ascending aorta and part of the superior vena cava's course. Left ventricular long-axis and short-axis views are challenging to display the terminal portion of the superior vena cava to the right atrium, and the suprasternal aortic arch long-axis view, while it may show part of the superior vena cava's course in some patients, is difficult to cover the lower segment of the superior vena cava to the right atrium in most cases. Clinically, the *Parasternal Short Axis View*, *Apical Four Chamber View*, and *Subxiphoid Biatrial View* are commonly used to observe the right atrium from different angles. We found that in the supine position during catheter placement, most patients had a good acoustic window for the *Apical Four Chamber View*, allowing a better observation of

**Table 4. Operation time of IC-ECG and TTE (FAS).**

|  | N | MEANA±STD | 95%CI | MIN~MAX | t | P |
|---|---|---|---|---|---|---|
| **IC-ECG** | 35(0) | 2.7±0.7 | 2.46~2.91 | 2.0~4.0 | 5.663[a] | P < 0.05 |
| **TTE** | 35(0) | 4.3±1.6 | 3.83~4.86 | 2.0~7.0 |  |  |

SD, standard deviation.

[a]t-Test for paired comparison.

**Table 5. Pain Score of TTE.**

|  | N | MEANA±STD | 95%CI | MIN~MAX |
|---|---|---|---|---|
| **TTE** | 35(0) | 2.1±1.1 | 1.77~2.49 | 1.0~4.0 |

the catheter tip. However, the visibility of the *Parasternal Short Axis View* was lower, possibly due to the patient lying flat during catheter placement, which prevents complete left lateral positioning and leads to a poorer acoustic window. The subxiphoid biatrial view could clearly show the CAJ and accurately measure the length of the catheter entering the right atrium. Technically, it is an ideal viewing angle. However, this viewing angle is only clear in patients with a thin and long body shape and a vertically positioned heart, with a relatively low visibility rate of the catheter tip in the majority of the population.

## 4.2  TTE localization technique offers a high degree of consistency with ECG

All patients in this study underwent routine postoperative chest X-ray to confirm catheter position; however, chest radiography provides only approximate localization and is not regarded as a true gold standard. For this reason, IC-ECG was selected as the primary reference standard. We acknowledge this as a limitation and recommend that future studies incorporate more accurate modalities, such as transesophageal echocardiography (TEE) or digital subtraction angiography (DSA), to provide more rigorous validation.

In this study, comparison of the mean catheter tip depth between TTE and ECG showed no statistically significant difference (Table 3). However, simple hypothesis testing is insufficient to demonstrate methodological equivalence, because a non-significant result may merely reflect limited sample size. To further assess agreement, Bland–Altman analysis was performed. The per-protocol analysis (Fig 2) demonstrated a minimal mean bias (−0.13) with narrow 95% limits of agreement (−1.10 to 0.83), indicating good concordance between the two techniques without systematic bias. Considering the five patients in whom TTE measurements were not obtainable, an intention-to-diagnose analysis was conducted using extreme value imputation. Even under this worst-case scenario (Fig 3), the mean bias (−0.34) and 95% limits of agreement (−1.70 to 1.00) remained within clinically acceptable ranges. Collectively, these results demonstrate a high degree of consistency between TTE and ECG in catheter tip localization.

Corradi et al. recently reported that contrast-enhanced TTE [12], using the epigastric bicaval acoustic view, achieved high sensitivity and specificity for catheter tip localization when validated against TEE. While this work highlighted the diagnostic potential of ultrasound-based methods, it was limited to a contrast-enhanced approach and a specific surgical population. By contrast, our study examined non-contrast TTE and validated its performance against IC-ECG, which is more broadly applied in daily clinical practice as a radiation-free standard. This provides a new and complementary perspective: non-contrast TTE offers good consistency with IC-ECG, short operation times, and minimal patient discomfort, while avoiding the need for contrast injection. Furthermore, the present study proposes a standardized methodological framework for non-contrast TTE, which has not been systematically described in previous reports. Taken together, these findings represent a novel contribution by demonstrating that non-contrast TTE may serve as a feasible and practical alternative, particularly in patients where IC-ECG cannot be applied (e.g., atrial fibrillation or absent P-waves).

## 4.3  The TTE technique requires more time to localize the catheter tip

The operation time of the ECG group was $2.7 \pm 0.7$ minutes, which was significantly lower than that of the TTE group ($4.3 \pm 1.6$ minutes). This may be related to the skillfulness of the operator. In this study, the cannula operator had extensive experience in cannula placement and skillful ECG technique. As for the application of TTE technology, this is an exploratory study. This is the first time that the Center has attempted to use TTE to locate the tip of a catheter. In order not to violate the principle of sterility, the operator must be very careful. In some patients, it is difficult to visualize the corresponding section because of the thick fat layer and the position of the patient. Although TTE required more time than IC-ECG, the difference was relatively small, and both procedures could be completed rapidly in routine practice. The slightly longer duration for TTE may be attributable to the operator's learning curve and the additional imaging steps.

### 4.4  TTE techniques can cause some degree of pain

Most patients experienced only mild pain during probe compression, but the pain is acceptable. The ECG technique was considered painless and in order to avoid patient confusion about the pain caused by the puncture, pain scores caused by the ECG technique were not counted in this study.

### 4.5  Complications

After the patients were implanted with central venous access, the study group continued to follow them for one week. There were no complications. Our biggest concern was the development of infection, as the ultrasound probe needs to be close to the chest wall. Throughout the operation, we tried to ensure a maximal sterile barrier.

### 4.6  Limitations

This study has several limitations. It was conducted in a single centre by a single highly experienced operator, which may limit external validity and preclude assessment of operator learning curves. The sample size was relatively small and did not allow for formal equivalence or non-inferiority testing; therefore, the agreement estimates should be interpreted as exploratory. TTE visualization of the catheter tip was not feasible in 5/35 patients (14%), indicating that patient habitus and acoustic windows may restrict applicability in unselected populations. Finally, the cohort consisted solely of oncology patients, and the results may not generalize to other clinical settings.

## Conclusion

Within these constraints, the present findings suggest that TTE may be feasible for PICC tip confirmation in selected oncology patients, with preliminary agreement between TTE and IC-ECG in cases with successful visualization. However, the non-visualization rate and the single-centre, single-operator design limit generalizability. Larger multicentre studies involving multiple operators and broader patient populations are needed to validate performance and define the clinical role of TTE in this setting.

## Supporting information

**S1 Data. Minimal dataset.** An anonymized dataset supporting the findings of this study.
(ZIP)

## Acknowledgments

We thank all participants in this study.

## Author contributions

**Conceptualization:** Ling Yuan.

**Formal analysis:** Kai Xin.

**Investigation:** Shanping Li, Chenjuan Wang, Lin Wang.

**Supervision:** Baorui Liu.

**Writing – original draft:** Kai Xin, Mian Zhou, Yubiao Kang.

**Writing – review & editing:** Mian Zhou.

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
