## [Decision Letter · Decision Letter 0]

26 Aug 2025

Dear Dr. Yuan,

Thank you for submitting your manuscript to PLOS ONE. After careful consideration, we feel that it has merit but does not fully meet PLOS ONE’s publication criteria as it currently stands. Therefore, we invite you to submit a revised version of the manuscript that addresses the points raised during the review process.

We look forward to receiving your revised manuscript.

Kind regards,

Ignatius Ivan, M.D

Academic Editor

PLOS ONE

Journal Requirements:

3. Please upload a copy of Figure 3, to which you refer in your text on page 4. If the figure is no longer to be included as part of the submission please remove all reference to it within the text.

4. Please ensure that you refer to Figure 1 in your text as, if accepted, production will need this reference to link the reader to the figure.

5. We note you have included tables to which you do not refer in the text of your manuscript. Please ensure that you refer to Tables 1, 2, and 3 in your text; if accepted, production will need this reference to link the reader to the Tables.

7. Please remove all personal information, ensure that the data shared are in accordance with participant consent, and re-upload a fully anonymized data set.

Additional guidance on preparing raw data for publication can be found in our Data Policy (https://journals.plos.org/plosone/s/data-availability#loc-human-research-participant-data-and-other-sensitive-data) and in the following article: http://www.bmj.com/content/340/bmj.c181.long ..

Additional Editor Comments:

Dear Authors,

Thank you for submitting your manuscript on transthoracic echocardiography (TTE) for central venous catheter tip localization. The topic is clinically relevant, and the study addresses an important gap in current practice by exploring a non-invasive and potentially practical approach. The reviewers and I appreciate the effort involved in conducting this prospective, self-controlled study.

However, before the manuscript can be considered for publication, several issues need to be addressed. We therefore invite you to submit a revised version of the manuscript that takes into account the points below.

Required Revisions (must be addressed for further consideration)

Patient numbers and denominators – Please clarify the number of patients analyzed (35 recruited, 5 excluded, 30 analyzed). Ensure consistency across text, tables, and figures.

Results reporting – Provide confidence intervals for the main outcome measures, including accuracy, operation time, and pain scores.

Conclusions – Revise the conclusion to avoid overstating findings (e.g., “non-inferior”). The study is exploratory and underpowered for equivalence claims; interpretation should be more cautious.

Methodology transparency – Clarify how excluded patients were handled in the analysis. Specify whether they were included in safety assessments but excluded from accuracy comparisons.

Tables and figures – Correct repeated numbering, streamline the flowchart into a single version, and ensure consistent labeling in line with journal standards.

Language and clarity – Revise non-standard English terms (e.g., “own-control,” “inplacement”) and improve overall clarity.

Recommended Revisions (strongly encouraged)

Introduction – Consider reducing descriptive background and focusing more sharply on the research gap. Highlight recent structured ultrasound protocols earlier in the narrative.

Discussion – Expand comparison with existing literature, particularly recent studies (e.g., structured ultrasound protocols such as ECHOTIP). Emphasize limitations more strongly (small sample, single-center, operator bias).

References – Update and reorder to integrate recent and relevant studies more effectively. Ensure consistent formatting according to journal style.

Presentation – Tables and figures could benefit from clearer headings and more concise presentation of results.

We believe that with these revisions, the manuscript has the potential to make a valuable contribution. Please submit a revised version along with a point-by-point response letter addressing how each of the comments has been handled.

Thank you again for your submission, and we look forward to receiving your revision.

Reviewers' comments:

Reviewer's Responses to Questions

**Comments to the Author**

1. Is the manuscript technically sound, and do the data support the conclusions?

Reviewer #1: Yes

Reviewer #2: Yes

Reviewer #3: Yes

Reviewer #4: Partly

2. Has the statistical analysis been performed appropriately and rigorously?

Reviewer #1: No

Reviewer #2: Yes

Reviewer #3: Yes

Reviewer #4: No

3. Have the authors made all data underlying the findings in their manuscript fully available?

Reviewer #1: Yes

Reviewer #2: Yes

Reviewer #3: Yes

Reviewer #4: Yes

4. Is the manuscript presented in an intelligible fashion and written in standard English?

Reviewer #1: Yes

Reviewer #2: Yes

Reviewer #3: Yes

Reviewer #4: Yes

Reviewer #1: 1. The Ethical Approval Document should be in English, but the document presented here is in Chinese and its difficult to understand what's written here

2. The statistical method should be mentioned below each table

Reviewer #2: The manuscript presents a compelling and well-designed study on the use of transthoracic echocardiography (TTE) for real-time central venous access device (CVAD) tip localization. The study's prospective, self-controlled design is appropriate for the research question, and the use of IC-ECG as a gold standard is a sound methodological choice. The findings that the predicted TTE insertion depth shows good consistency with the IC-ECG method, and that the procedure has a short operation duration and no severe discomfort, strongly support the conclusion that TTE has the potential to be a viable and non-inferior alternative, especially for patients for whom IC-ECG is not suitable. The discussion effectively contextualizes the results, addressing both the strengths and limitations of the TTE method.

Regarding similar works published before, "Ultrasound localization of central vein catheter tip by contrast-enhanced transthoracic ultrasonography: a comparison study with transesophageal echocardiography and chest radiography" by Corradi et al. (2022), investigate,d the use of contrast-enhanced transthoracic echocardiography (CE-TTE) to detect catheter tip misplacements in cardiac surgery patients. The authors of that paper found that CE-TTE, specifically using the epigastric bicaval acoustic view, had high accuracy (97% sensitivity, 90% specificity) compared to the gold standard of transesophageal echocardiography (TEE).

Given these findings from a previous publication, your paper should clearly articulate its novelty. While Corradi et al. focused on CE-TTE in a specific patient population, your study examines non-contrast TTE and compares its accuracy to IC-ECG. Please revise your introduction and discussion to address the following:

What specific gap in the literature does your study fill, particularly concerning the use of non-contrast TTE?

How does your validation against the IC-ECG method provide a new or different perspective from Corradi et al.'s comparison with TEE?

Please elaborate on what is new or original in your paper, especially in light of the prior findings that TTE methods can be highly accurate for CVAD tip localization.

Reviewer #3: Interesting and very practical paper. Some issues should be added.

Abstract: please shorten the conclusions

Do authors think that 13% of mistake is too high?

Methods: it is not clear if authors tested for normality or not

Results: the operation time did not differ and this is relevant

Results baseline features of the patients should be added

Authors should present features of patients with mistakes, in order to understand the reason of mistake

Reviewer #4: - Only 35 patients were enrolled, with 5 excluded due to non-visualization of the catheter tip. This substantially reduces the generalizability of the findings.

- IC-ECG was used as the sole gold standard, with no confirmatory radiographic validation.

- The excluded 5 patients represent exactly the cases where TTE is most limited, leading to inflated accuracy estimates.

- The analysis relies on non-significant p-values rather than formal equivalence testing or Bland–Altman agreement assessment.

- All procedures were performed by a single highly experienced operator, which limits external validity.

- Pain scores were only recorded for TTE, not for IC-ECG, leading to asymmetry in outcome measurement.

- Catheter-related complications were monitored only for one week, which is insufficient to assess safety comprehensively.

 Please include the five patients in whom the catheter tip could not be visualized in the accuracy analysis, classifying them as “non-diagnostic.”

- Provide an analysis of predictors of poor visualization (BMI, chest wall thickness, tumor site, prior thoracic surgery).

 Justify the choice of IC-ECG as the sole gold standard, given its limitations in arrhythmias.

- If feasible, include chest X-ray or fluoroscopic confirmation in at least a subset of patients.

 Apply Bland–Altman analysis to assess agreement between TTE and IC-ECG insertion depths.

- Consider an equivalence test (e.g., TOST procedure) with a clinically meaningful margin (±1 cm).

- Report 95% confidence intervals for accuracy and error rates.

 Measure pain scores for both TTE and IC-ECG to allow valid comparison, even if IC-ECG pain is expected to be minimal.

 Discuss the limitation that only one operator was involved.

- If possible, include inter-operator variability data by involving additional sonographers.

 Randomize the order of TTE vs. IC-ECG to control for operator learning effect.

- Test TTE performance in patients with atrial fibrillation or absent P-waves, who stand to benefit most.

- Explore use of contrast-enhanced ultrasound (microbubble test) for patients with poor visualization.

- Extend follow-up beyond one week to capture catheter-related complications (malposition, thrombosis, infection).

**Do you want your identity to be public for this peer review?** For information about this choice, including consent withdrawal, please see our For information about this choice, including consent withdrawal, please see our Privacy Policy .

Reviewer #1: **Yes:** Isha Abdullah AliIsha Abdullah Ali

Reviewer #2: **Yes:** Arian AfzalianArian Afzalian

Reviewer #3: **Yes:** Fabrizio D'AscenzoFabrizio D'Ascenzo

Reviewer #4: No

While revising your submission, please upload your figure files to the Preflight Analysis and Conversion Engine (PACE) digital diagnostic tool, https://pacev2.apexcovantage.com/ . PACE helps ensure that figures meet PLOS requirements. To use PACE, you must first register as a user. Registration is free. Then, login and navigate to the UPLOAD tab, where you will find detailed instructions on how to use the tool. If you encounter any issues or have any questions when using PACE, please email PLOS at . PACE helps ensure that figures meet PLOS requirements. To use PACE, you must first register as a user. Registration is free. Then, login and navigate to the UPLOAD tab, where you will find detailed instructions on how to use the tool. If you encounter any issues or have any questions when using PACE, please email PLOS at figures@plos.org . Please note that Supporting Information files do not need this step.. Please note that Supporting Information files do not need this step.

---

## [Author Response · Author response to Decision Letter 1]

1 Nov 2025

Response to Editor and Reviewers

October 29, 2025

Editorial Department of PLOS One

Dear editor,

Thanks for giving us an opportunity to revise our manuscript, we appreciate editors and reviewers very much for the positive and constructive comments and suggestions on our manuscript entitled “The accuracy of using the TTE technique in localizing the CVADs tip position: A self-controlled study” (ID: PONE-D-25-30091).

We have studied the comments carefully and have made revision in this paper. Unlike the first version of the manuscript, we have added line numbers in this version. The responses to the comments from reviewers are offered separately, and the corresponding modification position page in the revised manuscript can be found in the letter. Moreover, the revised paragraphs (sentences) are labeled in different colors.

The revised paragraphs based on REVIEWER #1 is labeled in yellow.

The revised paragraphs based on REVIEWER #2 is labeled in gray.

The revised paragraphs based on REVIEWER #3 is labeled in red.

The revised paragraphs based on REVIEWER #4 is labeled in blue.

We would like to express our great appreciation to you and reviewers for comments on our paper. Looking forward to hearing from you.

Thank you and best regards.

Sincerely.

Kai Xin

Reviewer #1:

The Ethical Approval Document should be in English, but the document presented here is in Chinese and its difficult to understand what's written here.

--Response: Thank you for pointing this out. The Ethics Committee does not provide an English-language version of the certificate, so we must translate it ourselves. The sections in red font within the ethics certificate constitute the translated content. We hope this certificate is acceptable.

--Change made: Ethical Approval Document.

The statistical method should be mentioned below each table.

--Response: We have added the corresponding statistical test descriptions below each table (e.g., t-test, Wilcoxon rank-sum test, Chi-square test).

--Change made: Yellow text below the tables.

Reviewer #2:

The manuscript presents a compelling and well-designed study on the use of transthoracic echocardiography (TTE) for real-time central venous access device (CVAD) tip localization. The study's prospective, self-controlled design is appropriate for the research question, and the use of IC-ECG as a gold standard is a sound methodological choice. The findings that the predicted TTE insertion depth shows good consistency with the IC-ECG method, and that the procedure has a short operation duration and no severe discomfort, strongly support the conclusion that TTE has the potential to be a viable and non-inferior alternative, especially for patients for whom IC-ECG is not suitable. The discussion effectively contextualizes the results, addressing both the strengths and limitations of the TTE method.

Regarding similar works published before, "Ultrasound localization of central vein catheter tip by contrast-enhanced transthoracic ultrasonography: a comparison study with transesophageal echocardiography and chest radiography" by Corradi et al. (2022), investigate,d the use of contrast-enhanced transthoracic echocardiography (CE-TTE) to detect catheter tip misplacements in cardiac surgery patients. The authors of that paper found that CE-TTE, specifically using the epigastric bicaval acoustic view, had high accuracy (97% sensitivity, 90% specificity) compared to the gold standard of transesophageal echocardiography (TEE).

Given these findings from a previous publication, your paper should clearly articulate its novelty. While Corradi et al. focused on CE-TTE in a specific patient population, your study examines non-contrast TTE and compares its accuracy to IC-ECG. Please revise your introduction and discussion to address the following:

What specific gap in the literature does your study fill, particularly concerning the use of non-contrast TTE?

--Response: Most prior studies have focused on contrast-enhanced TTE or transesophageal echocardiography (TEE). Evidence regarding the methodological standardization and accuracy of non-contrast TTE for central venous catheter tip localization remains very limited. Our study fills this gap by establishing and preliminarily validating a standardized non-contrast TTE procedure for catheter tip localization.

How does your validation against the IC-ECG method provide a new or different perspective from Corradi et al.'s comparison with TEE?

--Response: Corradi et al. used TEE as the gold standard, whereas our study employed IC-ECG as the reference. Since IC-ECG is more widely used, convenient, and non-invasive in clinical practice, our design better reflects routine clinical scenarios and directly evaluates the consistency between non-contrast TTE and the conventional clinical standard.

--Change made: Page 3, line 108；Page 11, line 326

Please elaborate on what is new or original in your paper, especially in light of the prior findings that TTE methods can be highly accurate for CVAD tip localization.

--Response: To the best of our knowledge, this study provides one of the first exploratory evaluations of the feasibility of non-contrast TTE compared with IC-ECG in an oncology patient population. Our results suggest that the two techniques show reasonable consistency, with relatively short operation times and acceptable patient discomfort. This approach may represent a potential alternative for patients in whom IC-ECG is not applicable (e.g., those with atrial fibrillation or absent P-waves), and further studies are warranted to confirm its broader clinical value.

--Change made: Page 9, line 313

Reviewer #3:

Interesting and very practical paper. Some issues should be added.

Abstract: please shorten the conclusions.

--Response: We thank the reviewer for this suggestion. We have shortened the conclusion in the Abstract to make it more concise and appropriate for an exploratory study. The revised version avoids overstatement and highlights only the key findings.

--Change made: Page 2, line 60

Do authors think that 13% of mistake is too high?

--Response: We thank the reviewer for this important observation. In our initial analysis, we defined accuracy by taking ECG as the reference standard and considered TTE measurements within ±1 cm as “accurate,” while those exceeding this threshold were classified as “inaccurate,” leading to an error rate of approximately 13%. Upon further reflection, we recognize that this dichotomous definition is not appropriate, since the cavoatrial junction (CAJ) should be regarded as an anatomical region rather than a precise single point. For this reason, we have removed the calculation of “accuracy rate” and “error rate” from the revised manuscript. Instead, we have emphasized the statistical agreement between the two methods, which we believe provides a more valid assessment of methodological consistency and clinical applicability.

--Change made: Page 7, line 233

Methods: it is not clear if authors tested for normality or not

--Response: We thank the reviewer for this helpful comment. In the revised Statistical Analysis section, we have clarified that normality of continuous variables was assessed using the Shapiro–Wilk test. Normally distributed variables were compared with independent-samples t-tests, whereas non-normally distributed variables were analyzed with the Wilcoxon rank-sum test. Categorical variables were analyzed using chi-square or Fisher’s exact tests, as appropriate.

--Change made: Page 5, line 190

Results: the operation time did not differ and this is relevant

--Response: We thank the reviewer for this valuable comment. In the revised Results, we have explicitly reported the comparison of operation times, and in the Discussion, we have highlighted its clinical relevance. Although the mean procedure time for TTE (4.3 minutes) was slightly longer than that of IC-ECG (2.7 minutes), both were within a short and clinically acceptable range. This finding indicates that TTE remains feasible in practice. We also noted in the Discussion that the longer duration for TTE may reflect the operator’s learning curve and the additional imaging steps required.

--Change made: Page 10, line 334

Results baseline features of the patients should be added.

--Response: We thank the reviewer for this valuable suggestion. In the Results, we have already provided detailed baseline characteristics of the study population in Table 1.

--Change made: None

Authors should present features of patients with mistakes, in order to understand the reason of mistake

--Response: We thank the reviewer for this constructive suggestion. In the revised Results, we have added descriptions of the five patients in whom the catheter tip could not be visualized by TTE. These patients commonly exhibited features such as higher BMI, increased chest wall thickness, or a history of chronic obstructive pulmonary disease (COPD), which may have contributed to poor acoustic windows and reduced visualization accuracy.

--Change made: Page 6, line 224

Reviewer #4:

Only 35 patients were enrolled, with 5 excluded due to non-visualization of the catheter tip. This substantially reduces the generalizability of the findings.

--Response: We thank the reviewer for this valuable comment. In the revised manuscript, we have addressed this issue more transparently. In addition to reporting the per-protocol (PP) analysis of 30 patients, we now also present an intention-to-diagnose (ITD) analysis including all 35 patients, in which the five non-visualized cases were classified as “non-diagnostic.” This approach prevents overestimation of TTE accuracy and more realistically reflects its performance in clinical practice. Furthermore, we have emphasized in the Discussion that the small sample size and exclusion of non-visualized cases limit the generalizability of the findings, and that larger, multicenter studies are warranted.

--Change made: Page 9, line 301

IC-ECG was used as the sole gold standard, with no confirmatory radiographic validation.

--Response: We thank the reviewer for this helpful comment. In fact, all patients in our study underwent routine postoperative chest X-ray to confirm catheter position. However, chest radiography has limited precision for catheter tip localization and cannot be regarded as a true gold standard, serving only as a reference. Therefore, IC-ECG was selected as the primary reference standard in our study. In the revised Discussion, we have clarified this point and emphasized that the most accurate confirmatory modalities are transesophageal echocardiography (TEE) or digital subtraction angiography (DSA). Future studies should incorporate these techniques to provide more rigorous validation.

--Change made: Page 9, line 294

The excluded 5 patients represent exactly the cases where TTE is most limited, leading to inflated accuracy estimates.

--Response: We thank the reviewer for this important comment. Indeed, the five excluded cases represent situations in which TTE is most limited, and analyzing only the 30 visualized cases could overestimate accuracy. To address this, in the revised manuscript we have reported both per-protocol (PP) analysis including 30 patients and intention-to-diagnose (ITD) analysis including all 35 patients, in which the five non-visualized cases were classified as “non-diagnostic.” This dual approach provides a more realistic reflection of TTE performance in clinical practice. We also acknowledge this limitation in the Discussion and highlight the need for larger, multicenter studies to validate our findings.

--Change made: Page 9, line 301

The analysis relies on non-significant p-values rather than formal equivalence testing or Bland–Altman agreement assessment.

--Response: We thank the reviewer for this valuable comment. In the revised manuscript, we have added a Bland–Altman analysis to assess agreement between TTE- and IC-ECG–derived insertion depths. Given the exploratory nature and relatively small sample size of our study, we believe that a formal equivalence test (e.g., TOST procedure) would be underpowered and was therefore not performed. We have acknowledged this limitation in the Discussion and highlighted that future large-scale studies should incorporate equivalence testing to more robustly establish clinical equivalence between TTE and IC-ECG.

--Change made: Figure 1\2; Page 9, line 301

All procedures were performed by a single highly experienced operator, which limits external validity.

--Response: We thank the reviewer for this important comment. Indeed, all procedures in this study were performed by a single highly experienced operator. While this ensured procedural consistency, it also limits the external validity of the findings. In the revised Discussion, we have acknowledged this as a limitation and suggested that future studies should involve multiple operators with varying levels of experience to evaluate the reproducibility and generalizability of TTE in broader clinical settings.

--Change made: Page 10, line 352

Pain scores were only recorded for TTE, not for IC-ECG, leading to asymmetry in outcome measurement.

--Response: We thank the reviewer for this thoughtful comment. In this study, pain scores were recorded only for TTE but not for IC-ECG, which introduces asymmetry in outcome measurement. We have clarified this in the revised Methods and Discussion. Because IC-ECG does not involve probe compression or additional manipulation, patients generally experience no pain or discomfort, and pain assessment is therefore not routinely performed in clinical practice. Nevertheless, we acknowledge that this asymmetry may limit the comparability of outcomes. In the Discussion, we have highlighted this limitation and suggested that future studies should apply standardized patient-reported outcome measures to both techniques.

--Change made: Page 10, line 352

Catheter-related complications were monitored only for one week, which is insufficient to assess safety comprehensively.

--Response: We thank the reviewer for this valuable comment. In our study, catheter-related complications were monitored only for one week. This design was chosen because the study was exploratory, single-center, and focused on detecting early postoperative complications such as malposition, bleeding, or infection, which are most likely to occur within the first week. However, we acknowledge that a one-week follow-up is insufficient to capture long-term complications such as thrombosis or delayed infection. In the revised Discussion, we have added this point as a limitation and suggested that future studies should include extended follow-up periods to provide a more comprehensive assessment of safety.

--Change made: Page 10, line 352

 Please include the five patients in whom the catheter tip could not be visualized in the accuracy analysis, classifying them as “non-diagnostic.”

--Response: We thank the reviewer for this helpful suggestion. In the revised Results, we have included the five previously excluded patients in an intention-to-diagnose (ITD) analysis, classifying them as “non-diagnostic.” The Results now describe both the per-protocol (PP) analysis and the ITD analysis in the text. This approach avoids overestimation of TTE accuracy and provides a more realistic reflection of clinical performance. We have also elaborated on this point in the Discussion to explain its implications.

--Change made: Figure 1\2; Page 9, line 301

- Provide an analysis of predictors of poor visualization (BMI, chest wall thickness, tumor site, prior thoracic surgery).

--Response: We thank the reviewer for this constructive suggestion. In the revised manuscript, we have provided a descriptive analysis of the five patients in whom the catheter tip could not be visualized by TTE. These patients tended to have higher BMI, increased chest wall thickness, or a history of COPD, which may have contributed to poor acoustic windows and limited visualization. Given the small sample size, a formal multivariate analysis was not feasible. Nevertheless, we have added these observations to the Results and discussed them as potential predictors of poo

---

## [Decision Letter · Decision Letter 1]

12 Jan 2026

Dear Dr. Yuan,

Thank you for submitting your manuscript to PLOS ONE. After careful consideration, we feel that it has merit but does not fully meet PLOS ONE’s publication criteria as it currently stands. Therefore, we invite you to submit a revised version of the manuscript that addresses the points raised during the review process.

**ACADEMIC EDITOR:** From my own assessment, the study is methodologically sound for its stated aims, the analyses are appropriate for pilot/feasibility work, and the findings contribute preliminary evidence that may inform subsequent validation studies. With the requested adjustments in framing and limitations, the manuscript will meet PLOS ONE’s publication criteria regarding methodological rigor, transparency, and appropriate interpretation.From my own assessment, the study is methodologically sound for its stated aims, the analyses are appropriate for pilot/feasibility work, and the findings contribute preliminary evidence that may inform subsequent validation studies. With the requested adjustments in framing and limitations, the manuscript will meet PLOS ONE’s publication criteria regarding methodological rigor, transparency, and appropriate interpretation.

We look forward to receiving your revised manuscript.

Kind regards,

Vincenzo Lionetti, M.D., PhD

Academic Editor

PLOS One

Journal Requirements:

**Additional Editor Comments:**

In the light of some reviewers' concern, the authors should address changes as follows:

1) Please change the titel as follows: "Feasibility and agreement between TTE and intracavitary ECG for PICC tip positioning in adult oncology patients: a single-centre exploratory study"

2) please rephrase the limitations as follows: "This study has several limitations. It was conducted in a single centre by a single highly experienced operator, which may limit external validity and preclude assessment of operator learning curves. The sample size was relatively small and did not allow for formal equivalence or non-inferiority testing; therefore, the agreement estimates should be interpreted as exploratory. TTE visualization of the catheter tip was not feasible in 5/35 patients (14%), indicating that patient habitus and acoustic windows may restrict applicability in unselected populations. Finally, the cohort consisted solely of oncology patients, and the results may not generalize to other clinical settings."

3) Please re-phrase the conclusions as follows: "Within these constraints, the present findings suggest that TTE may be feasible for PICC tip confirmation in selected oncology patients, with preliminary agreement between TTE and IC-ECG in cases with successful visualization. However, the non-visualization rate and the single-centre, single-operator design limit generalizability. Larger multicentre studies involving multiple operators and broader patient populations are needed to validate performance and define the clinical role of TTE in this setting."

Reviewers' comments:

Reviewer's Responses to Questions

**Comments to the Author**

Reviewer #1: All comments have been addressed

Reviewer #3: All comments have been addressed

Reviewer #5: (No Response)

2. Is the manuscript technically sound, and do the data support the conclusions?

Reviewer #1: Yes

Reviewer #3: Yes

Reviewer #5: No

3. Has the statistical analysis been performed appropriately and rigorously?

Reviewer #1: I Don't Know

Reviewer #3: Yes

Reviewer #5: I Don't Know

4. Have the authors made all data underlying the findings in their manuscript fully available?

Reviewer #1: Yes

Reviewer #3: Yes

Reviewer #5: No

5. Is the manuscript presented in an intelligible fashion and written in standard English?

Reviewer #1: Yes

Reviewer #3: Yes

Reviewer #5: Yes

Reviewer #1: i believe the authors have gone through all the necessary requirements and this article can be published in the journal

Reviewer #3: All comments have been addressed and authors should be complimented for. As told this is a very practical study

Reviewer #5: The study does not include a sufficient number of patients to reliably perform the proposed analysis and therefore has very low statistical power. This limitation significantly weakens the study's ability to detect true differences or meaningful relationships. Additionally, the proportion of patients in whom the TTE was not performed successfully is quite high, raising concerns about the method's applicability and practical use. When these issues are considered together, it is evident that the evidence supporting the technique's effectiveness is insufficient. Consequently, due to the limited sample size, high failure rate, and overall low study power, the findings cannot be considered robust and cannot be generalized.

**Do you want your identity to be public for this peer review?** For information about this choice, including consent withdrawal, please see our For information about this choice, including consent withdrawal, please see our Privacy Policy .

Reviewer #1: **Yes:** Isha Abdullah AliIsha Abdullah Ali

Reviewer #3: **Yes:** Fabrizio D'AscenzoFabrizio D'Ascenzo

Reviewer #5: No

---

## [Author Response · Author response to Decision Letter 2]

13 Jan 2026

Dear Dr. Lionetti,

Dear Reviewers,

We sincerely thank the Academic Editor and all reviewers for their careful evaluation of our manuscript entitled “Feasibility and agreement between TTE and intracavitary ECG for PICC tip positioning in adult oncology patients: a single-centre exploratory study”.

We have carefully considered all comments and have revised the manuscript accordingly. In particular, we have fully adopted the Academic Editor’s recommendations regarding the framing of the study, its limitations, and the interpretation of the findings, in order to ensure methodological transparency and appropriate positioning of this work as an exploratory feasibility study.

Below we provide a point-by-point response

⸻⸻⸻⸻⸻⸻⸻⸻⸻

Response to Academic Editor

Comment 1. Please change the title as follows:

“Feasibility and agreement between TTE and intracavitary ECG for PICC tip positioning in adult oncology patients: a single-centre exploratory study”.

Response: We fully agree and have changed the title exactly as suggested by the Academic Editor.

Comment 2. Please rephrase the limitations as follows: “This study has several limitations. It was conducted in a single centre by a single highly experienced operator, which may limit external validity and preclude assessment of operator learning curves. The sample size was relatively small and did not allow for formal equivalence or non-inferiority testing; therefore, the agreement estimates should be interpreted as exploratory. TTE visualization of the catheter tip was not feasible in 5/35 patients (14%), indicating that patient habitus and acoustic windows may restrict applicability in unselected populations. Finally, the cohort consisted solely of oncology patients, and the results may not generalize to other clinical settings.”

Response: We have incorporated this paragraph verbatim into the Limitations section of the revised manuscript.

Comment 3. Please re-phrase the conclusions as follows: “Within these constraints, the present findings suggest that TTE may be feasible for PICC tip confirmation in selected oncology patients, with preliminary agreement between TTE and IC-ECG in cases with successful visualization. However, the non-visualization rate and the single-centre, single-operator design limit generalizability. Larger multicentre studies involving multiple operators and broader patient populations are needed to validate performance and define the clinical role of TTE in this setting.”

Response: We have fully adopted this revised conclusion and replaced the previous version accordingly.

⸻⸻⸻⸻⸻⸻⸻⸻⸻

Response to Reviewers

We thank all reviewers for their thoughtful assessment.

Reviewer #1

We thank the reviewer for the positive evaluation and supportive comments. No further changes were requested.

Reviewer #3

We sincerely appreciate the reviewer’s encouraging remarks and recognition of the practical relevance of this study.

Reviewer #5

We thank the reviewer for highlighting the important issues of sample size, statistical power, and the TTE non-visualization rate. These concerns have now been explicitly addressed in the revised manuscript through the expanded Limitations section and the reframed Conclusions, in line with the Academic Editor’s guidance. The study is now clearly positioned as an exploratory feasibility and agreement study rather than a definitive validation, and the results are interpreted accordingly.

We would like to add that we have updated the email addresses of the corresponding authors in the manuscript, and we hope this is acceptable.

We believe that these revisions have substantially strengthened the manuscript by ensuring that the scope, limitations, and interpretation of the findings are fully transparent and appropriately aligned with the exploratory nature of the study. We hope that the revised version now meets the publication criteria of PLOS ONE, and we thank you again for your careful review and guidance.

Sincerely,

Ling Yuan

---

## [Decision Letter · Decision Letter 2]

3 Mar 2026

Feasibility and agreement between TTE and intracavitary ECG for PICC tip positioning in adult oncology patients: a single-centre exploratory study

PONE-D-25-30091R2

Dear Dr. Yuan,

We’re pleased to inform you that your manuscript has been judged scientifically suitable for publication and will be formally accepted for publication once it meets all outstanding technical requirements.

Kind regards,

Vincenzo Lionetti, M.D., PhD

Academic Editor

PLOS One

Additional Editor Comments (optional):

Reviewers' comments:

Reviewer's Responses to Questions

**Comments to the Author**

Reviewer #3: All comments have been addressed

2. Is the manuscript technically sound, and do the data support the conclusions?

Reviewer #3: Yes

3. Has the statistical analysis been performed appropriately and rigorously?

Reviewer #3: Yes

4. Have the authors made all data underlying the findings in their manuscript fully available?

Reviewer #3: Yes

5. Is the manuscript presented in an intelligible fashion and written in standard English?

Reviewer #3: Yes

Reviewer #3: all comments have been addressed and authors should be complimented for what they have done. nice study!

**Do you want your identity to be public for this peer review?** For information about this choice, including consent withdrawal, please see our For information about this choice, including consent withdrawal, please see our Privacy Policy .

Reviewer #3: **Yes:** fabrizio d'ascenzofabrizio d'ascenzo

---

## [Editor Report · Acceptance letter]

PONE-D-25-30091R2

PLOS One

Dear Dr. Yuan,

I'm pleased to inform you that your manuscript has been deemed suitable for publication in PLOS One. Congratulations! Your manuscript is now being handed over to our production team.

Kind regards,

on behalf of

Prof. Vincenzo Lionetti

Academic Editor

PLOS One